# The challenges of fieldwork: Improving the experience for women in coastal sciences

Sarah M. Hamylton[1] ⓘ, Hannah E. Power[2], Shari L. Gallop[3,4] and Ana Vila-Concejo[5] ⓘ

[1]School of Earth, Atmospheric and Life Sciences, University of Wollongong, Wollongong, NSW, Australia; [2]School of Environmental and Life Sciences, University of Newcastle, Callaghan, NSW, Australia; [3]School of Science, University of Waikato, Waikato, New Zealand; [4]Pattle Delamore Partners Ltd, Tauranga, New Zealand and [5]Geocoastal Research Group, Marine Studies Institute, School of Geosciences, University of Sydney, Sydney, NSW, Australia

## Perspective

**Keywords:**
beach; coast; human activity; gender discrimination; fieldwork; fieldtrip

**Corresponding author:**
Sarah M. Hamylton;
Email: shamylto@uow.edu.au

## Abstract

Women face disproportionate challenges while undertaking coastal fieldwork. We draw on 18 responses that specifically raise fieldwork issues from an international survey about perceptions and experiences of gender inequality for those working in coastal sciences to discuss two common themes. These themes are barriers to fieldwork participation and challenges for women working in coastal field settings such as boats or working on beaches, including discrimination and sexual harassment. We suggest five priority behavioural and policy changes to improve the fieldwork experience for women in coastal sciences: (i) publicise field role models and trail blazers, (ii) improve opportunities and capacity for women to undertake fieldwork, (iii) establish field codes of conduct, (iv) acknowledge the challenges women face in the field and provide support where possible, and (v) foster an enjoyable and supportive fieldwork culture.

## Impact statement

The specific challenges that women face while undertaking fieldwork in coastal environments are identified from a survey of 314 coastal scientists. These include difficulties getting into the field through selective invitations, competing responsibilities and lack of facilities for women at field sites and onboard scientific boats. Under representation in field settings, as well as reconfigured social boundaries, work environments and sleeping arrangements expose women to vulnerable situations, discrimination and sexual harassment. Suggestions for improvement include publicising women in the field as role models, improving opportunities for and the capacity for women to undertake fieldwork, establishing codes of behavioural conduct for the field, acknowledging challenges and providing specific support where possible and fostering an enjoyable and supportive fieldwork culture.

## Introduction

Fieldwork provides a critical opportunity to gather environmental data, inspire emerging scientists, develop skills, expand networks and participate in collaborative research. Yet surveys reveal that many women experience disproportionate challenges in the field (Clancy et al., 2014). Surveys of coastal scientists and engineers reveal that the fieldwork-related challenges for women are multifaceted, including lack of fieldwork-active female role models, remote and urban coastal settings that are unsafe for women, limited capacity to participate in fieldtrips, gender stereotyping in the field and discriminatory assumptions about women's ability to perform fieldwork tasks. These challenges can begin before reaching the field and can raise unique issues for women (Vila-Concejo et al., 2018; Adams et al., 2020; Clair, 2021).

The last few years have seen an increased awareness of the challenges that women field scientists face, with corresponding efforts to understand and address women's experiences of fieldwork in ocean and coastal sciences (Brooks and Déniz-González, 2021; Hill et al., 2021; Kelly and Yarincik, 2021). The present paper draws on survey results from the Women in Coastal Geosciences and Engineering (WICGE) network to consolidate issues experienced by women specifically undertaking coastal fieldwork into two themes for further discussion. These two themes are (i) barriers to fieldwork participation (envisaging the possibility of fieldwork, opportunities for inclusion in fieldtrips), and (ii) specific challenges for women in a coastal setting. We conclude by outlining five practical suggestions for improving the fieldwork experience for women in coastal sciences.

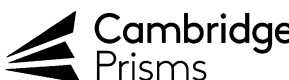

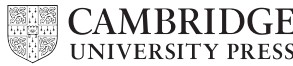

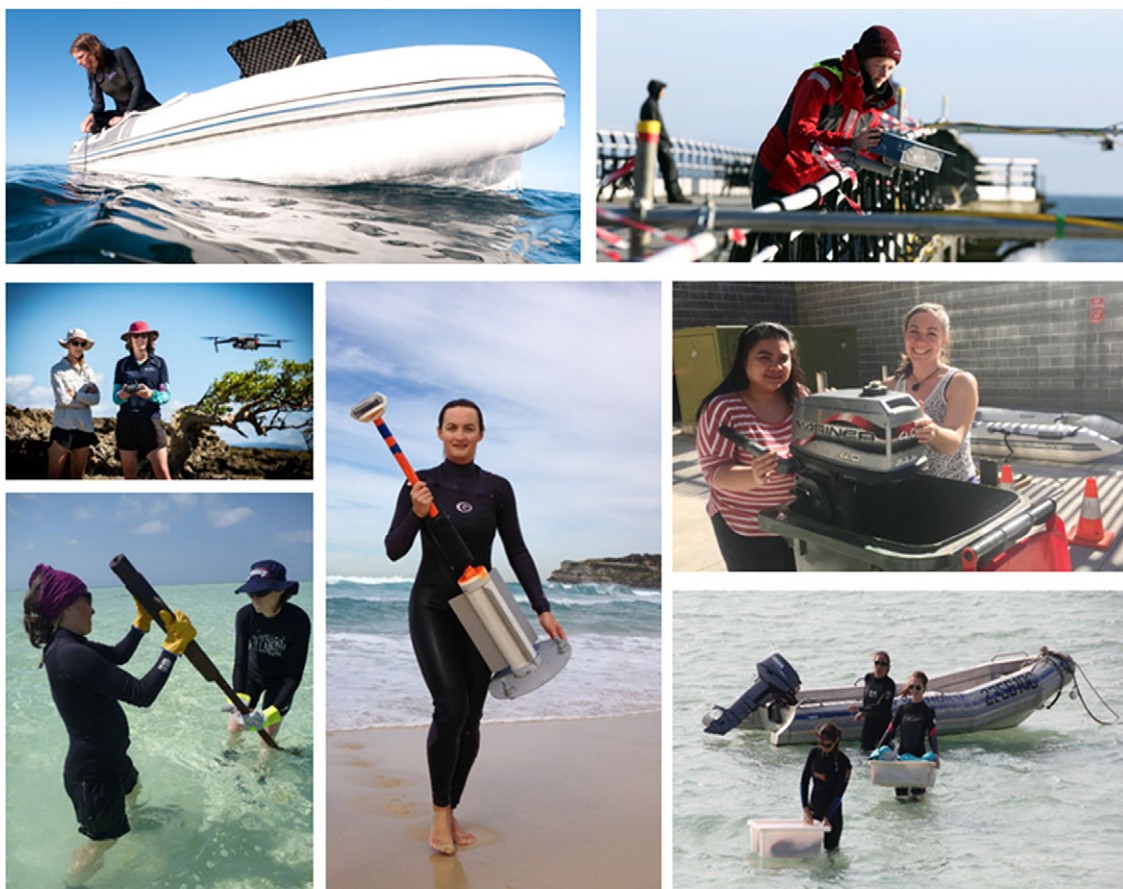

**Figure 1.** Disrupting the narrative: Women fieldworkers operating equipment, carrying gear and fixing engines.

## Approach

We present responses from an international survey about perceptions and experiences of gender inequality for those working in coastal sciences. A paper-based survey was initially launched during the 14th International Coastal Symposium (Sydney, Australia in March 2016). Further responses were solicited via an online questionnaire that was posted on the WICGE website and circulated through social media channels to assess experiences of gender equality for those working in coastal sciences (Vila-Concejo et al., 2018). Here, we draw on a subset of survey responses pertaining to issues faced while undertaking fieldwork to frame a discussion of the emerging issues faced by women undertaking fieldwork in coastal sciences. Specifically, all survey responses were evaluated and answers were extracted that mentioned issues relating to fieldwork in response to the question: *"If you are willing to do so, please provide a short description of the gender inequality you have directly experienced or observed while working as a coastal geoscientist and/ or engineer".*

To protect the welfare and rights of all research participants, the questionnaire and associated materials were assessed for integrity by the Social Sciences Human Research Ethics Committee at the University of Wollongong. The full questionnaire and responses were deposited in the Dataverse repository (doi: 10.7910/DVN/F1B2FS).

## Synthesis

The survey yielded 314 responses overall (34% male, 65% female and 1% other). Over 95% of these respondents were coastal geoscience and engineering (CGE) professionals working as university researchers, government scientists and industry consultants. Of the 314 responses received, 113 respondents provided examples of gender inequality that they had either directly experienced or observed while working in coastal sciences. Overall, 36% of survey respondents therefore supplied examples of gender inequality that they had experienced or observed and within these responses and 16% of overall respondents provided examples that related to fieldwork. Box 1 outlines direct quotes from 18 respondents regarding fieldwork-related issues that emerged when participants were asked to describe gender inequality experienced while working in coastal sciences.

These responses highlight two common themes, which we discuss below: (i) barriers to fieldwork participation (envisaging the possibility of fieldwork, opportunities for inclusion in field-trips), and (ii) challenges in the field (sexual harassment and specific challenges for women in a coastal setting).

Women face barriers to participation in fieldtrips. Their fieldwork abilities are commonly underrated and women undergraduate university students are less likely to consider themselves fit for fieldwork (Maguire, 1998). Such beliefs can be shaped by symbolic portrayals in magazines, online videos, on social media and in degree promotional brochures that disproportionately depict coastal scientists as being white, physically fit males. Such portrayals signal to those outside the profession what is possible in the field and a lack of visible role models makes it difficult for those

**Box 1.** Issues faced by survey respondents while undertaking fieldwork.

*If you are willing to do so, please provide a short description of the gender inequality that you have directly experienced or observed while working as a coastal geoscientist and / or engineer.*

- Colleagues preferring men over women in allocation of tasks ranging from fieldwork through to management *(man, senior career, university researcher).*
- Sometimes women are "advised" to avoid field works, for security reasons (or they are considered weak, or we are threaten by rape for being with a lot of men) *(woman, early career, university researcher).*
- I was told that a male colleague would be better to coordinate and lead a fieldtrip *(woman, early career, university researcher).*
- As I fill in this survey, the corridor of the building I work in is lined with empty offices. My colleagues are out on boats doing fieldwork. I have a passion for coastal science. That's why I'm working in a university. But I have a disproportionately large share of administrative, pastoral and governance duties that keep me from engaging in my passion. I'm about to go to a committee meeting of women, doing women's work (reviewing teaching offerings). Inequality is alive and well in my workplace! *(woman, mid-career, university researcher).*
- I've twice experienced harassment on fieldwork expeditions *(woman, early career, university researcher).*
- During a field campaign I was treated inequal compared to male colleagues by one of the team members. He tried to be helpful by doing jobs that he deemed too hard (in terms of lifting equipment) for me, but it was very irritating *(woman, mid-career, university researcher).*
- Inability to do field work because of religious values, no women on boats during Ramadan *(woman, mid-career, university researcher).*
- Not allowed to join research vessels *(woman, early career, research institute).*
- Only woman doing fieldwork. Never worked for a female boss. Rarely worked with female peers. When applying for an internal role that involved travel told by a male "why would a mother want to apply for a role that involves travel"? Lack of role models/mentors/peers/female colleagues *(woman, mid-career, university researcher).*
- Prevented from research in the field because of gender *(woman, mid-career, government).*
- During fieldwork, as a woman, I am not included in tasks that are considered more male oriented like heavy lifting or being helpful while deploying instruments. I try to make myself included but I keep getting passed over for the nearest male (whom is not closer than me) *(woman, early career, government).*
- For physical disparity field capacity (carrying heavy loads, prejudices) *(woman, mid-career, other).*
- Opportunities to participate in field work have preferentially been given to men *(woman, early career, government).*
- Inequality is inherent as we have to care for our family. Having had three children in the last 3 years, I had to go on maternity leave, which slowed down by publishing, and I haven't been able to go on fieldwork, cruises or even conferences. However, it was my decision to have kids and I knew what the result would be. Men stay more free, even when they have kids, but it's not their fault! *(woman, mid-career, university researcher).*
- Being asked to help with outreach on a field experiment rather than setup of equipment, having setup of field equipment checked more frequently that it was done correctly than male counterparts *(woman, early career, university researcher).*
- Many examples I've seen (and I'm male!), here are a few (and none are exaggerated). (1) Saying we can only appoint males to field roles as women are too weak to pick stuff up. (2) Having staff expect their female students to act as baby sitters (male students are never asked). (3) Female students being told "you can only leave my supervision if you become a lesbian" (4) obvious misogyny at conferences – for example, [*Name of senior Professor*]'s use of a female swimsuit model to give examples of different beach modelling approaches (a highly embarrassing, but not unexpected thing to have been said). (5) Not account taken of child rearing in appointments panels. (6) Numerous comments on female students looks (7) females only been selected for short listing to make it look like it is gender balanced, with no intention of them being appointed. "We need an extra woman on this list for the Faculty" (8) staff yelling "I want to be an amateur gynaecologist" as female students hand in assignments *(man, mid-career, university researcher).*

- I have also observed female students and staff being left out of field experience for "not being strong enough" *(man, senior career, university researcher).*
- I have experienced inequality in a very direct, blatant manner – for example, I was banned from a fieldtrip to collect information at one of PhD research sites in Saudi Arabia. I have had my ideas ignored in meetings (then subsequently listened to when repeated by male colleagues). Probably, the starkest inequality I have experienced is that I am not able to work the same (extended) hours or conduct fieldwork in the same manner as my male colleagues while managing a family at home *(woman, mid-career, university researcher).*
- When field work includes a boat travels *(woman, mid-career, university researcher).*

falling outside this narrow remit to envision themselves as coastal scientists (Mol and Atchison, 2019).

Women's participation in coastal fieldtrips may be prevented via outright bans of women joining trips or being allowed on scientific cruises, selective invitations or competing responsibilities. These responsibilities might include a disproportionate share of academic teaching and governance work in an educational institution, or caring for children and elderly family members at home, all of which often fall onto the shoulders of women, precluding participation in extended fieldtrips (Vila-Concejo et al., 2018).

Coastal field settings raise unique challenges for women. For example, women typically represent a very small proportion of people working from boats or in remote field camps, where personal space is reduced and fieldworkers can be required to sleep in close proximity, potentially exposing women to vulnerable situations. Women can face inadequate facilities at sea for toileting, managing menstruation or lactation (Orcutt and Cetinić, 2014). While women are in the minority and the social boundaries that characterise everyday working life are reconfigured (e.g. when working from a boat), women coastal scientists are at greater exposure to microaggressions, discrimination, abuse and sexual harassment. Fieldwork attire for working around water such as close-fitting wetsuits and swimsuits may increase the likelihood of womens' bodies being objectified by colleagues. Further, the interconnected nature of multiple aspects of identity including race, religion, class and sexuality, can create overlapping and intersectional disadvantages for fieldwork-active women (Núñez et al., 2020), which are beyond the scope of the current analysis.

## Suggestions for improvement

Changes must be made to improve the field experience for women. Recent workshops and reports have advanced our appreciation of the scope and diversity of issues faced, leading to recommendations for addressing these issues at institutional and individual levels (Johnson et al., 2018; Kelly, 2021). There have been steps forward in relation to codes of conduct that outline acceptable behaviour, are integrated into relevant existing departments, policies and procedures (e.g., fieldwork safety guidelines and bystander training for witnesses of sexual assault) and provide reporting structures to facilitate resolution of complaints. These are becoming more common in fieldwork-active institutions such as engineering and environmental consultancies, research groups and universities (see the Royal Geographical Society's webpage on principles for safe, responsible and ethical fieldwork for illustrative codes of conduct at https://www.rgs.org/research/higher-education-resources/field principle3/). Such codes inspire a welcoming and supportive

behavioural culture in the field and encourage those experiencing challenges to speak up.

To encourage best practice in future coastal research, we suggest five priority behavioural and policy changes to improve the field-work experience for women in coastal sciences:

1. *Publicise field role models and trail blazers:* Fieldtrip leaders and others promoting fieldwork should develop representational material to reshape public views of coastal scientists in the field, emphasising fieldwork possibilities for women by increasing the visibility of their participation and offering counter-narratives to address gendered stereotypes (e.g., Figure 1).

2. *Improve opportunities and capacity for women to undertake fieldwork:* Trip organisers should strive for diverse field teams by identifying and addressing the intersecting disadvantages experienced by women. This may include, for example, making provision for other responsibilities that arise during the period of fieldwork, including professional duties and family-related care.

3. *Establish field codes of conduct:* Fieldwork codes should outline acceptable standards of behaviour on fieldtrips, what constitutes misconduct, sexual harassment and assault, how to make a complaint and disciplinary measures in the event of misconduct.

4. *Acknowledge the challenges women face in the field and provide support where possible:* Prior to entering the field, fieldwork leaders should include a briefing for all participants that explicitly acknowledges practical challenges that may arise for women in remote locations and outlines how these have been addressed, including managing toileting and menstruation with the provision of pop-up toilet facilities, private areas or breaks where possible.

5. *Foster an enjoyable and supportive fieldwork culture:* Emphasize mutual respect, safety, inclusivity and collegiality on every fieldtrip. Regularly check in on the welfare of members of the field team individually, providing the means for fieldworkers to communicate with family, such as via satellite phones with pre-agreed usage agreements negotiated on a case by case basis, while working in the field (Thomas et al., 2003).

By discussing some of the challenges faced by women coastal scientists and offering suggestions to address these, we hope to provoke constructive conversations that improve the fieldwork experience for women, both in coastal settings and across the broader geosciences.

**Open peer review.** To view the open peer review materials for this article, please visit http://doi.org/10.1017/cft.2023.26.

**Data availability statement.** A copy of the survey is outlined in Table S4 in the Supplementary information of Vila Concejo et al. (https://static-content. springer.com/esm/art%3A10.1057%2Fs41599-018-0154-0/MediaObjects/ 41599_2018_154_MOESM1_ESM.docx). Survey responses were deposited in the Harvard Dataverse repository (see https://dataverse.harvard.edu/dataset. xhtml?persistentId=doi:10.7910/DVN/F1B2FS).

**Acknowledgements.** We acknowledge that the challenges women face in the field involve intersecting aspects of identity including, but not limited to, race, religion, class, sexuality and gender identity. A detailed discussion of each of these components of identity is beyond the scope of the current paper. We thank all who responded to the WICGE survey. This work would not have been possible without the work of the WICGE Committee, including Karin Bryan, Luciana Esteves, Graziela Miot da Silva, Amaia Ruiz de Alegria Arzaburu, Nadia Senechal, Emilia Guisado, Irene Delgado-Fernandez, Kristen Splinter, Naomi Edwards, Siddhi Joshi, Astrid Blom and Rose Palermo.

**Author contribution.** All authors contributed to project design and gathering of primary data of coastal sciences professionals, discussion of data analysis and writing this paper. S.H. drafted the paper. H.P., S.G. and A.V.C. helped to author the paper, including reviewing field challenges specific to coastal sciences and devising suggestions for improvement.

**Financial support.** This research did not receive any specific grant from funding agencies in the public, commercial, or not-for-profit sectors.

**Competing interest.** The authors declare no competing interests exist.

**Ethics statement.** The survey was approved by the University of Wollongong Social Sciences Human Research Ethics Committee (Ethics Number 2016/052).

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
