## [Reviewer Report]

I would like to commend the authors on tackling and publishing on this topic. I highly agree that it is important to study the challenges that women face in coastal field work and I agree with many of the suggestions and comments stated. Nevertheless, I recommend major revisions. This recommendation is based on the following concerns:

- Quotes and survey results were extracted from a subset of responses. It is unclear how this subset was chosen and is defined, and if bias is introduced through it.

- From the current manuscript, it is not possible to assess how representative the results are of the community as a whole, and are there also positive examples for programs, investments, and improvement? I understand that the discussed issues should not occur at all, but this question is relevant since it also provides a measure of if solutions or improvements are in place in parts of the community and how they specifically could look like.

- Recommendations are very general and maybe not surprising. A great contribution would be to discuss actual guidelines. For example, the example of the satellite phone is mentioned, but I believe many and most expeditions will carry this technology with them to-date. The bigger question is what is a usage agreement that is reasonable and appropriate: should there be one rule or case-by-case agreements? how would they look like? And how is access managed? Another example is related to boat facilities. How can this be added if at the same time vessels may be restricted in size for accessibility of sites? Or should there be a maximum time on vessel before landing with access to facilities are provided? Many of these solutions will take significant financial investments that will unlikely happen within a short time or maybe at all. What can be solutions in the mean time? What are acceptable improvements?

---

## [Reviewer Report]

Authors: Hamylton, Power, Gallop, Vila-Concejo

The challenges of fieldwork: Improving the experience for women in coastal sciences.

This is an important, timely contribution that investigates the fieldwork experiences for woman in coastal sciences. Inputs were provided from participants of the 14th International Coastal Symposium and from a questionnaire posted on the Women In Coastal Geosciences and Engineering website (total of 314 responses). The suggestions for improvement are well described based on the study surveys, recent workshops and reports. This perspective article is well written, and I only found two typos i.e. Line 11, challenges, no caps and Line 88, others.

The authors acknowledge that “the challenges women face in the field involve intersecting aspects of identity including, but not limited to, race, religion, class, sexuality and gender identity.” A limitation of this study is that no details were provided on this in the article; of the 314 responses how many different countries were represented, how many different races, religions?

In South Africa, we have safety and security issues and for coastal field work we prefer a male present. Rape and murder are a threat in several urban as well as remote areas resulting in us no longer sampling some estuary sites or abandoning night zooplankton sampling because too dangerous. “Threats to a vibrant research community include a paucity of local funding and growing safety issues for field ecologists. Safety of field researchers is a particular problem in KwaZulu-Natal, but also in urban environments around the coast, with research and monitoring activities having been curtailed in some areas”. (Adams et al. 2020, DOI: 10.2989/16085914.2020.1751980).

---

## [Reviewer Report]

I enjoyed reading this commentary on a survey undertaken by WICGE. However, the importance of this paper extends beyond my enjoyment - it is critical that discussions of women undertaking coastal fieldwork be discussed with a broader audience than women if change is to occur. And this change is crucial given the growing contribution of women to coastal geoscience and engineering.

Below I have outlined some suggestions that may improve the manuscript:

1. The manuscript is written in the traditional scientific format (intro, methods, results and discussion), and I found that this left me wanting to see what analyses of the survey were being undertaken. In addition, the methods did not detail the survey questions that preceeded the opportunity to comment. Personally - I do not think this is problemmatic as the survey comments should be made publicly available, but to manage reader expectations I think it would be better to present the paper as a commentary rather than in the traditional scientific paper format. I suggest that the best way to address this is to change the headings (e.g. Methods  Approach, Results and discissions - Synthesis). I acknowledge that the capacity to adjust headings may be dependent on the author guidelines for this journal.

2. As the paper is focussed on fieldwork experiences in coastal research, I would have liked to see either more connection to the existing literature regarding fieldwork in geosciences (there is a lot of information emerging), and/or more explicit discussion about why women undertaking coastal fieldwork may need attention beyond what is undertaken more broadly in the field disciplines of geoscience and engineering. I suspect that the latter is more suited to this journal. I am aware that there are hints at this in the comments (e.g. boating, heavy equipment). I can see this being addressed with an additional couple of sentences in the introduction that includes citations to the existing research on geoscience fieldwork, and emphasising that the survey results allow for the unique experiences in coastal research to be highlighted.

3. The authors have provided a great list of suggestions, but I suggest linking through to some of the existing ‘good’ examples of codes of conduct (there are a few already available online). The Times higher Education did a great article on this a few years ago on this topic (https://www.timeshighereducation.com/campus/how-develop-code-conduct-ethical-research-fieldwork).

4. I was a bit rankled reading that ‘women should be briefed on practical challenges that may arise’. From experience, women are often more aware than men of the challenges of remote fieldwork, and I suspect men may need some briefing. This could be as simple as developing approaches for toileting (where, frequency, no-questions asked) that is shared with everyone, emphasising inclusive conduct on boats and when using heavy equipment that is conveyed to everyone.

Line 88: ‘other’ should be changed to ‘others’

Thanks for the opportunity to read this manuscript - it was insighful.

---

## [Editor Report]

Dear Authors,

Thank you for submitting this Perspective piece to Coastal Futures. As you can see, you have received three reviews on the submission, all of which are very favourable of your work but also raise some points I would like you to consider. More transparency/detail on the data used would be quite helpful, and 2 reviewers also suggested that discussing what is already in place/codes/etc would also be good and I’d tend to agree. 

I appreciate you might be quite limited by space, but perhaps this is something that can be discussed as I think it’s important to add that bit of depth to this work to increase its impact even more. 

I look forward to the revised version. 

Kristen Splinter

Handling Senior Editor, Coastal Futures

---

## [Reviewer Report]

I appreciated the work presented in this paper. The authors have also taken care to address my suggestions and comments. I can also see that the authros ahve addressed otehr reviewer comments and the paper has now been improved in a waythat reflects my previous suggestion of ‘minor revisions’. I am very happy with how the authros addressed all reviewer comments.

---

## [Reviewer Report]

I would like to thank the authors for the revisions. I believe most of my previously stated questions and concerns have been addressed. One question remains for me that may also be of interest to other readers: Of the 314 overall responses, do the 17 respondents represented in Box 1 represent all responses to this question (ie, 297 respondents did not provide a short description of the gender inequality experienced/observed) or is this a selection? Would this suggest that 5% of the respondents are willing/able to share experiences of gender inequality in coastal field work, and if not, how many of the total respondents reported having experienced/observed gender inequality during coastal field work?

---

## [Editor Report]

R2 has asked for I think a very interesting clarification if it would be great to address this before considering the paper for publication.

---

## [Reviewer Report]

Thank you. I believe my comments have been well addressed.

Thank you for tackling this important topic.

---

## [Editor Report]

I think the paper is a worthy contribution to the conversation on gender equity within the Coastal field and should be published. I noted a few sentences in my final reading that could use your attention I feel. 

L60: “ Specifically, all survey responses were evaluated we extracted any answers that mentioned issues” - there is something missing in this sentence. 

L76: “ 18 respondents regarding fieldwork-related issues that emerged from seventeen responses” - is it 18 or 17 or am I missing something?

Thanks kindly,

Kristen Splinter

Senior Editor, Cambridge Prisms: Coastal Futures.